# CSS: Contrastive Semantic Similarity for Uncertainty Quantification of LLMs

Shuang Ao[1]                    Stefan Rueger[2]                    Advaith Siddharthan[3]

[1]Knowledge Media Institute (KMi), The Open University, Milton Keynes, UK

## Abstract

Despite the impressive capability of large language models (LLMs), knowing when to trust their generations remains an open challenge. The recent literature on uncertainty quantification of natural language generation (NLG) utilizes a conventional natural language inference (NLI) classifier to measure the semantic dispersion of LLMs responses. These studies employ logits of NLI classifier for semantic clustering to estimate uncertainty. However, logits represent the probability of the predicted class and barely contain feature information for potential clustering. Alternatively, CLIP (Contrastive Language–Image Pre-training) performs impressively in extracting image-text pair features and measuring their similarity. To extend its usability, we propose Contrastive Semantic Similarity, the CLIP-based feature extraction module to obtain similarity features for measuring uncertainty for text pairs. We apply this method to selective NLG, which detects and rejects unreliable generations for better trustworthiness of LLMs. We conduct extensive experiments with three LLMs on several benchmark question-answering datasets with comprehensive evaluation metrics. Results show that our proposed method performs better in estimating reliable responses of LLMs than comparable baselines. The code are available at `https://github.com/AoShuang92/css_uq_llms`.

## 1 INTRODUCTION

Despite recent breakthroughs in a wide range of natural language generation (NLG) tasks [Hoffmann et al., 2022, Touvron et al., 2023, Chowdhery et al., 2023], the uncertainty quantification (UQ) of large language models (LLMs)

remains an open challenge. Without reliable measures of uncertainty, it is implausible to apply LLMs in critical tasks such as medical [Singhal et al., 2023] or legal question-answering [Louis et al., 2023], or medical diagnosing [Wang et al., 2023]. A reliable measure of uncertainty helps to decide when to trust a model, which is also the key problem in building safer AI systems [Hendrycks et al., 2021]. Recently, LLMs have been deployed in the industry as powerful tools to assist professional or personal work, with well-known interfaces such as ChatGPT[1], Gemini[2] and Perplexity AI[3]. However, with the enhanced capabilities of LLMs, concerns are simultaneously raised about their trustworthiness.

The study of UQ in LLMs has gained significant attention recently. Most existing methods are white-box, relying on either calculating entropy from predicted probabilities [Malinin and Gales, 2020, Kuhn et al., 2023] or querying models for their prediction confidence [Lin et al., 2022b, Kadavath et al., 2022]. However, these techniques often require task-specific labels, additional training data, or white-box access to the internal model information. Black-box UQ strategies address this by analyzing the consistency of information across model generations. Techniques like n-gram overlap [Fomicheva et al., 2020] assess surface-level similarity, while more recent approaches explore semantic equivalence [Kuhn et al., 2023, Lin et al., 2023]. These methods cluster sentences based on meaning to estimate uncertainty, with a higher number of clusters indicating greater semantic diversity and thus higher LLM uncertainty. However, a key limitation lies in using Natural Language Inference (NLI) classifier logits to measure semantic equivalence. Logits represent class probabilities, not the semantic features needed for accurate clustering. This highlights the need for more sophisticated features that better capture the true semantic relationships between generated texts.

The Contrastive Language-Image Pre-training (CLIP) [Rad-

---

[1]`https://chat.openai.com/`
[2]`https://gemini.google.com/`
[3]`https://www.perplexity.ai/`

ford et al., 2021] learns the link between textual semantics and their visual representations rather than mapping features to a fixed set of predetermined object categories. In other words, it captures similarity features in a contrastive approach by learning how much a given text snippet relates to an image. Inspired by its promising function, we design CLIP to contrastively extract similarity features between text pairs, where semantic relations can be represented by feature patterns learned from the model. We propose the contrastive semantic similarity (CCS), where features contain implicit information about the semantic relations of text inputs. Our method allows the transitivity between the measurement of semantic equivalence and the inner semantic relations between text pairs. It also provides insightful clustering information to form semantic sets and further uncertainty estimation.

We evaluate our method with selective NLG [Ren et al., 2022, Cole et al., 2023], a self-assessment evaluation method to detect when the generations of LLMs are unreliable. Responses with high-uncertainty are likely to be wrongly generated, which will diminish the trustworthiness of a model. Therefore, accurate uncertainty estimation can provide higher performance in selective answering. The evaluation is conducted with the area under the accuracy/rejection trade-off curve. In this paper, we conduct extensive evaluation on several open and closed book free-form question answering benchmark datasets, with sampled set of answers for a given question generated by SOTA LLMs. Results show the superiority of our proposed method over the NLI classifier logits. Our contributions and findings are summarized as below:

1. We design a novel technique for UQ in LLMs that utilizes Contrastive Semantic Similarity (CSS) to extract insightful semantic relations between text pairs.

2. We modify the CLIP text encoder to obtain text-text pairs semantic similarities, then employ spectral clustering technique to estimate uncertainty of sampled generations of LLMs.

3. By conducting extensive experiments on LLMs and question-answering datasets, together with extensive ablation studies, we report:

   (a) our proposed method outperforms SOTA UQ techniques, indicating the contrastive semantic similarity contains more semantic information than NLI logits;

   (b) Contrastive feature extraction of CLIP are superior to regular language models, extending their application scope in language generation;

   (c) our proposed method enhances selective NLG by detecting unreliable generations more accurately, which reflect the effectiveness of our method for UQ in LLMs.

## 2 RELATED WORK

The study of UQ has attracted great attention in deep learning tasks such as classification or regression [Lakshminarayanan et al., 2017, Kendall and Gal, 2017, Abdar et al., 2021, Ao et al., 2023a]. However, most UQ techniques are not transferable to generative AI due to the unique challenges in free-form NLG in terms of (1) entropy calculation of the utmost high-dimension probability, (2) texts with distinct tokens but with identical meanings, and (3) accessibility of token-level probability or fine-tuning for end-users. To solve the extremely high-dimension output issue, [Malinin and Gales, 2020] utilize the geometric mean token-probability to calculate the length-normalizing predictive entropy, based on the prior empirical success of [Murray and Chiang, 2018]. Moreover, a recent study introduces a novel entropy-based uncertainty measure called semantic entropy [Kuhn et al., 2023], incorporating linguistic invariances created by shared meanings.

Word overlap metrics such as METEOR [Banerjee and Lavie, 2005], BLEU [Papineni et al., 2002] and ROUGE [Lin, 2004] are typically used to measure similarities between text pairs. However, distinct tokens may carry similar semantic meanings, and these methods may fail to extract semantic relations between text pairs. To highlight semantic meanings in free-from NLG, semantic equivalence [Kuhn et al., 2023] is introduced via the bi-directional entailment algorithm of natural language inference (NLI), which is further utilized to cluster generations of LLMs based on their semantic meanings. Utilizing the concept of entailment to measure the semantic relations between text pairs is logical and understandable from a linguistic perspective. In other words, two sentences are semantically equivalent if they entail each other. This novel method is a breakthrough for text clusters based on semantic meanings instead of traditional n-gram token counting, but it still requires access to predicted probability. To measure the uncertainty of LLMs in a post-hoc fashion, Graph Laplacian is employed to cluster LLM generations that are represented by the NLI classifier [Lin et al., 2023].

Selective NLG (also referred to as selective answering/generation, NLG with rejection) is the main application to evaluate the effectiveness of UQ methods for language generation. Samples with higher uncertainty are likely to be wrongly predicted or generated, and rejecting them can improve the reliability of the model. It is analogous to the commonly used term selective prediction in classification [Lin et al., 2022a, Geifman and El-Yaniv, 2017, Ao et al., 2023b]. Both tasks can determine when to trust a model, whether it is a classifier or an LLM. Selective answering benefits the decision-making process and improves the trustworthiness of LLMs by detecting their failure outputs.

# 3 METHODOLOGY

This section discusses uncertainty quantification methods for LLMs based on measuring the information consistency across $m$ generated responses $\{r_1, r_2, \ldots, r_m\}$ for a given input question $x$.

## 3.1 BACKGROUND

**NLI Classifier.** The natural language inference (NLI) classifier has been used to measure semantic similarities for text pairs. The NLI classifier predicts classes as entailment, neutral and contradiction, via utilizing the pre-trained off-the-shelf DeBERTa model [He et al., 2020].

**Semantic Entropy with NLI Classifier.** Semantic equivalence of text pairs can be measured with NLI classifier logits/scores (referred as NLI logits for simplification). The NLI logits denotes as $s_{r_i, r_j}$ for text pairs $r_i, r_j$. If two sentences can entail each other, they share similar semantic meanings. Based on this linguistic concept, the recent study [Kuhn et al., 2023] hereby introduces the bi-directional entailment algorithm to measure semantic similarity between text pairs. All generations are clustered into three semantic sets by the predicted label of NLI logits. To obtain the likelihood of a semantic set, the predicted probability of each sentence in the cluster is accumulated. Given $m$ sampled responses of a given question, larger semantic sets indicate higher information consistency or lower uncertainty, as more sentences carry similar meanings. With the given input $x$ and its corresponding sampled $m$ responses, suppose number of semantic clusters as $C$, the semantic entropy ($SE$) estimated by Monte Carlo integration is written as: $SE(x) \approx -|C|^{-1} \sum_{i=1}^{|C|} \log p(C_i \mid x)$. This method requires access to the predicted probabilities of LLMs. One limitation of this work lies in the over-simplified clustering, as ambiguous responses can belong to more than one class. Furthermore, the equivalence between NLI logits judged cluster and real semantic clusters is not guaranteed [Lin et al., 2023].

**Graph Laplacian with NLI Classifier.** Given the pairwise similarities represented by NLI logits $s_{r_i, r_j}$, but without obtaining predicted probabilities of each generation, a straightforward way to cluster $m$ generations is via spectral clustering. For an input question $x$, let $R = \{r_i\}_{i=1}^{m}$ be the generation set for each item as a node. Based on the bi-directional entailment algorithm, the semantic relations between $r_i, r_j$ defined by NLI logits is written as: $w_{i,j} = (s_{i,j} + s_{j,i})/2$. Hence the symmetric weighted adjacency matrix $W$ is $W = (w_{i,j})_{i,j=1,\ldots,m}$. The degree matrix $D$ is a diagonal matrix, where a node with a higher degree means well-connected with other nodes. The higher degree of one generation suggests it carries similar meanings with other generations, resulting in the lower uncertainty of

LLMs. The degree for $r_i, r_j$ is written as: $D_{ii} = \sum_j W_{ij}$. When there are semantic relations between $r_i$ and $r_j$, $i = j$ and $D_{ii}$ is non-zero; otherwise $D_{ij}$ is 0. The pairwise distance represents the semantic difference between text pairs, and the degree matrix to estimate uncertainty is written as:

$$U_{\text{Deg}} = \text{trace}(m - D)/m^2 \tag{1}$$

The graph Laplacian $L$ is thereby: $L := D - W$. The eigenvalues of $L$ are non-negative and sorted in ascending order: $\lambda_1 \leq \lambda_2 \leq \ldots \leq \lambda_n$. The eigenvectors form an orthogonal basis: $v_1, v_2 \ldots, v_n$.

The corresponding eigenvalues and eigenvectors are used to measure the uncertainty of the sampled generation set $R$. In spectral clustering, the distribution of eigenvalues is used to determine the number of clusters [Von Luxburg, 2007]. Under the context of uncertainty for LLMs, the multiplicity of the zero eigenvalues coincides with the number of semantic sets [Lin et al., 2023]. Thus the uncertainly estimated by eigenvalues-based semantic clusters ($U_{set}$) can be written as:

$$U_{\text{Eig}} = \sum_{k=1}^{m} \max(0, 1 - \lambda_k) \tag{2}$$

As the eigenvalue in graph Laplacian, $\lambda$ coincides with the number of semantic clusters. Following previous work [Lin et al., 2023, Von Luxburg, 2007], eigenvalues larger than 1 are ignored as only the smallest few eigenvalues carry important information about the clusters. Hence equation (2) picks the max value between 0 and $1 - \lambda_k$ to ignore eigenvalues larger than 1.

The eigenvectors are treated as coordinates for nodes (sampled generation). The informal embedding space $e_i$ for the generation $r_i$ can be formed as $\mathbf{e}_i = [v_{1,i}, \ldots, v_{n,i}]$ [Ng et al., 2001, Von Luxburg, 2007]. The average distance to the center is treated to measure uncertainty, named eccentricity ($U_{Ecc}$) which is written as:

$$U_{\text{Ecc}} = \left\| \left[ \mathbf{e}_1'^{\top}, \ldots, \mathbf{e}_m'^{\top} \right] \right\|_2 \tag{3}$$

in which $\mathbf{e}'$ demonstrate the offset from the average embedding. Eccentricity has been applied to measure uncertainty for LLMs in a black-box way [Lin et al., 2023], and also to detect out-of-distribution generations in conditional language models [Ren et al., 2022]. However, utilizing NLI logits to represent semantic similarities is still questionable as logits are only predicted probabilities. It is necessary to apply features that represent semantic relations for text pairs.

**Contrastive Feature Extraction.** CLIP is a contrastive approach to learn the link between textual and visual rep-

resentations, which is is trained on a large dataset of 400 million image-text pairs [Radford et al., 2021]. It learns a multi-modal embedding space from the transformer based image-encoder and text-encoder, where semantically similar images and texts are also similar in the joint embedding space. As a foundation model trained on vast amount of data, it has shown great capabilities in tasks such as language-driven image generation [Ramesh et al., 2022], zero-shot semantic segmentation [He et al., 2023] and text-guided image manipulation [Hou et al., 2022]. Hence utilizing CLIP to learn semantic similarities between text pairs can be a plausible approach.

## 3.2 UQ WITH CONTRASTIVE SEMANTIC SIMILARITIES

In this section, we propose Contrastive Semantic Similarities (CSS): the CLIP-based semantic similarity features for text pairs. We then utilize CSS in Graph Laplacian (GL) to estimate uncertainty for LLMs.

**Contrastive Semantic Similarities** Initially, CLIP learns the relation of text-image pairs via the jointly trained image and text encoder. By connecting images and texts in the same space, the cosine similarity of the embeddings for correct related image-text pairs is minimized and vice versa. To extract contrastive semantic similarity features, we solely utilize the text encoder for text-pairs embeddings, which avoids the discrepancy of multi-modal embeddings in joint space. For the text pair $r_i, r_j$, we first utilize CLIP text-encoder to extract features for each of them, then conduct point-wise product (Hadamard Product) on the corresponding embeddings to obtain similarity features, as demonstrated in Figure 1.

As CLIP is based on contrastive approach, the obtained features represent contrastive relations between text pairs, which is called Contrastive Semantic Similarities (CSS) in our work. CSS feature maps maintain the same dimension as embeddings. For better semantic clustering with graph Laplacian, we then apply principal component analysis (PCA) to reduce dimensions of CSS feature maps.

**Graph Laplacian with Contrastive Semantic Similarities** Let denote the CSS feature map for text-pair $r_i, r_j$ as $css_{r_i, r_j}$. Similar to estimating uncertainty with NLI logits, the symmetric weighted adjacency matrix for $R$ is $W^{css}$. $w_{i,j}^{css}$ is a scalar value obtained from the affinity matrix by projecting the similarity vector $css_{r_i, r_j}$. Suppose the degree matrix is $D^{css}$, the uncertainty for $m$ generations can be written as:

$$U_{\text{Deg}}^{css} = \text{trace}(m - D^{css})/m^2 \tag{4}$$

The graph Laplacian with CSS features then can be written

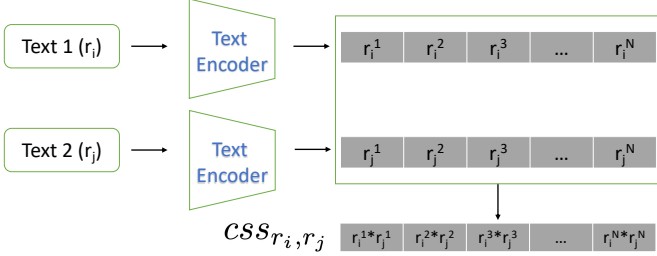

Figure 1: The demonstration of obtaining our proposed contrastive semantic similarities for text pairs. After passing each generation to the CLIP text encoder, we conduct point-wise product on the corresponding embeddings to obtain the similarity features.

as: $L^{css} := D^{css} - W^{css}$. The ascending order eigenvalues are $\lambda_1^{css} \le \lambda_2^{css} \le \ldots \le \lambda_n^{css}$, and corresponding eigenvectors are $v_1^{css}, v_2^{css} \ldots, v_n^{css}$. Recalled that eigenvalues represent number of semantic clusters, based on Eq. 2, the uncertainty $U_{set}^{css}$ is formed as:

$$U_{\text{Eig}}^{css} = \sum_{k=1}^{m} \max\left(0, 1 - \lambda_k^{css}\right) \tag{5}$$

The embedding space $e_i^{css}$ for generation $r_i$ now is formed with eigenvectors $v^{css}$ generated by similarity features. Given the offset the average embedding as $\mathbf{e}^{\mathbf{css}\prime}$, the eccentricity $U_{Ecc}^{css}$ as uncertainty is demonstrated as:

$$U_{\text{Ecc}}^{css} = \left\| \left[ \mathbf{e}^{\mathbf{css}\prime\top}_1, \ldots, \mathbf{e}^{\mathbf{css}\prime\top}_m \right] \right\|_2 \tag{6}$$

We applied PCA to reduce dimension of CSS feature maps for better clustering, with feature dimension of 64 in our experiments.

## 4 EXPERIMENTS

### 4.1 DATASET AND BASELINES

We use the open-book conversational question answering dataset CoQA [Reddy et al., 2019], a closed-book question answering dataset TriviaQA [Joshi et al., 2017], and a challenging closed-book QA dataset Natural Questions (NQ) [Kwiatkowski et al., 2019] for our experiments. We utilize the development/validation set for each dataset, respectively 7983, 9960 and 3610 samples for CoQA, TriviaQA and NQ. In terms of LLMs, LLaMA (with 13 billion parameters) [Touvron et al., 2023], OPT (with 13 billion parameters) [Zhang et al., 2022], and GPT (GPT-3.5-turbo) by OpenAI API are used to generate sampled responses for each question. For fair comparison, we use the official

implementation [4] [5] for all the baselines and we fixed the number of sampled generations of each question as $m = 20$.

We compare our proposed method with the following state of the art techniques:

1. Lexical Similarity (LexiSim) [Lin, 2004]: measures the average Rouge-L score among sampled generations.

2. Number of Semantically Distinct Answers (NumSem) [Kuhn et al., 2023]: leverages the count of semantically unique responses within correct and incorrect generations as a measure of uncertainty.

3. Semantic Entropy (SE) [Kuhn et al., 2023]: computes entropy over clusters formed by semantically equivalent samples, which required the access of token-level logits/predicted probabilities from LLMs.

4. P(true) [Kadavath et al., 2022]: estimates the probability of generations by querying the model itself if generations are true or false. This method utilizes the token-level logits, and we follow the experimental setup detailed in the originating study.

5. Graph Laplacian with NLI Classifier Logits (L-GL) [Lin et al., 2023]: demonstrates that semantic dispersion can effectively estimate the quality of generations of LLMs. By utilizing NLI logits to cluster generations with similar semantic meaning, the uncertainty is measured by invariances of GL, respectively eigenvalues (EigV), degree matrix (Deg), eigenvectors (Ecc).

### 4.2 IMPLEMENTATION DETAILS

For our experiments, we use the pre-trained CLIP model *openai/clip-vit-base-patch32* by using Huggingface library, which is trained on a dataset of about 400 million image-text pairs collected from the Internet. Our CSS takes about 2.3 seconds to calculate the UQ for a text pair, where previous work [Lin et al., 2023] takes about 1.2 seconds. This demonstrates a minor computational additional resource for our method, which is still quite fast. For all our experiments, we use the 2 GPUs of Nvidia Tesla P40 with 23 GB RAM. Generating 20 responses for each question takes about 30 - 50 seconds.

### 4.3 EVALUATION METRICS

Following the prior work of [Kuhn et al., 2023, Lin et al., 2023], we use the Rouge-L score and GPT correctness score as matching criteria to evaluate the correctness of generated responses. GPT correctness score is provided by *gpt-3.5-turbo* from the OpenAI API, which assigns a correctness

score between 0 and 1 for the similarity between given reference answer and generated responses. If the Rouge-L score for the generation and reference answer is larger than 0.3, the generation is considered to be correct. Similarly, the threshold for the GPT correctness score is 0.7.

To validate our proposed method in terms of selective answering, we apply Area Under Accuracy-Rejection Curve (AUARC) [Nadeem et al., 2009] as the evaluation metric. After applying baselines and our proposed method, each sample (one question with 20 sampled generations) obtains one score to represent the uncertainty. We rank all samples based on this score and reject higher-uncertainty ones to calculate accuracy for the remaining data. If the UQ method is effective and precise, samples with higher uncertainty are more likely to be wrongly predicted. Thus, the higher the AUARC, the better the quality of the UQ methods. To further examine the overall performance of LLMs, we follow previous works [Kuhn et al., 2023, Lin et al., 2023, Band et al., 2022] to employ Area Under Receiver Operating Characteristic (AUROC) to compare UQ methods. The uncertainty score for each sample serves as the threshold for calculating the sensitivity and specificity for the AUROC. A higher AUROC indicates lower uncertainty in LLMs, signifying that the sampled generations of a given question are more consistent.

## 5 RESULTS

Table 1 presents the AUARC results of sampled generations on the TriviaQA, CoQA, and NQ datasets using LLMs, with the Rouge-L score serving as the criterion for correctness. The results for white-box (WB) methods of semantic entropy (SE) and p(true) depend on token-level probabilities. As the ChatGPT API does not provide these, we are unable to report the corresponding AUARC and AUROC results.

When the model is perfectly calibrated, all rejected samples will be the wrong ones. In the table, Oracle represents this upper bound on AUARC performance.

As the only rule-based measurement (utilizing Rouge-L) among all methods, lexical similarity demonstrates a superior capability in estimating uncertainty compared to the Number of Semantically Distinct Answers (NumSem) in most cases. This suggests that variations in vocabulary or grammar contribute significantly to semantic meanings. For NLI logits-based methods, all three (labelled EigV, Ecc and Deg) sub-methods in NLI logits-based graph Laplacian perform better than semantic entropy in LLaMA-generated datasets.

The performance of our proposed CCS graph Laplacian is, on average, 1.5% to 2% higher than that of L-GL. CSS-Deg achieves better performance than CSS-EigV and CSS-Ecc in most cases.

---

[4]https://github.com/lorenzkuhn/semantic_uncertainty
[5]https://github.com/zlin7/UQ-NLG

Table 1: Results of AUARC with Rouge-L score as the correctness criterion, on sampled generation by LLaMA, OPT and GPT on dataset of TriviaQA, CoQA and NQ. Results of white-box (WB) methods semantic entropy (SE) and p(true) on GPT generation are not available. WB methods require the predicted probabilites of outputs, which are not provided in ChatGPT API. All results are shown in percentages for clarity. Best results are in bold for each dataset.

| Dataset | | TriviaQA | | | CoQA | | | NQ | | |
|---|---|---|---|---|---|---|---|---|---|---|
| Model | | LLaMA | OPT | GPT | LLaMA | OPT | GPT | LLaMA | OPT | GPT |
| | Acc | 57.57 | 25.60 | 81.07 | 55.96 | 51.99 | 66.38 | 19.32 | 9.10 | 39.83 |
| | Oracle | 89.60 | 54.30 | 97.91 | 85.10 | 78.56 | 93.00 | 42.35 | 24.15 | 75.58 |
| | NumSem | 73.25 | 33.76 | 81.07 | 64.31 | 57.29 | 67.81 | 20.85 | 10.58 | 45.97 |
| | LexiSim | 78.98 | 46.72 | 87.47 | 79.09 | **73.15** | 80.39 | 35.74 | 17.77 | 58.01 |
| L-GL | EigV | 80.67 | 48.70 | 92.32 | 79.54 | 71.96 | 84.34 | 33.58 | 14.72 | 62.13 |
| | Ecc | 80.20 | 48.83 | 92.01 | 78.92 | 70.96 | 83.94 | 34.26 | 17.41 | 61.80 |
| | Deg | 80.71 | 49.00 | 92.24 | 79.12 | 71.83 | 84.22 | 34.23 | 17.49 | 62.42 |
| WB | SE | 74.09 | 47.90 | – | 77.65 | 67.46 | – | 28.97 | 16.62 | – |
| | P(true) | 61.85 | 20.93 | – | 61.75 | 58.32 | – | 20.19 | 8.27 | – |
| Ours (CSS) | CSS-EigV | 81.47 | 49.85 | 92.70 | **81.92** | 72.13 | 87.26 | **36.80** | 18.10 | 64.83 |
| | CSS-Ecc | 81.29 | 49.60 | 93.07 | 80.83 | 71.36 | **87.34** | 36.62 | 18.19 | **65.04** |
| | CSS-Deg | **81.55** | **50.08** | **93.18** | 81.17 | 73.18 | 87.02 | 36.67 | **18.34** | 64.87 |

Table 2: Results of AUROC with Rouge-L score as the correctness criterion, on sampled generation by LLaMA, OPT and GPT on dataset of TriviaQA, CoQA and NQ. All results are shown in percentages for clarity. Best results are in bold for each dataset.

| Dataset | | TriviaQA | | | CoQA | | | NQ | | |
|---|---|---|---|---|---|---|---|---|---|---|
| Model | | LLaMA | OPT | GPT | LLaMA | OPT | GPT | LLaMA | OPT | GPT |
| | NumSem | 75.06 | 68.56 | 68.20 | 57.76 | 57.60 | 51.69 | 55.59 | 59.20 | 61.13 |
| | LexiSim | 77.63 | 76.48 | 81.13 | 75.72 | 76.40 | 68.70 | 76.72 | 73.90 | 71.65 |
| L-GL | EigV | 84.35 | 82.88 | **83.40** | 77.95 | 75.70 | 78.65 | 72.59 | 73.88 | 80.88 |
| | Ecc | 83.66 | 83.91 | 82.50 | 77.26 | 74.81 | 77.39 | 74.44 | 76.02 | 79.82 |
| | Deg | 84.52 | 83.36 | 82.93 | 77.53 | 75.85 | 78.76 | 74.01 | 74.75 | **81.31** |
| WB | SE | 74.39 | 81.54 | – | 74.55 | 71.25 | – | 69.50 | 74.61 | – |
| | P(true) | 55.12 | 41.64 | – | 55.14 | 52.67 | – | 52.52 | 47.92 | – |
| Ours (CSS) | CSS-EigV | 85.52 | 85.37 | 82.27 | **78.78** | **77.19** | 80.04 | **76.08** | **77.08** | 79.28 |
| | CSS-Ecc | 85.17 | 84.97 | 81.57 | 78.40 | 76.70 | **80.40** | 75.76 | 76.53 | 79.91 |
| | CSS-Deg | **85.63** | **85.82** | 81.77 | 78.68 | 76.95 | 79.12 | 75.81 | 77.25 | 80.01 |

The ARC depicted in Figure 2 illustrates how the eccentricity of our method (Ecc (ours)) outperforms other baselines in OPT-sampled generations for the CoQA dataset. As the rejection rate increases, our method demonstrates superior performance compared to other approaches, indicating improved uncertainty estimation through our contrastive technique. The AUROC results in table 2 are mostly consistent with AUARC results, where our proposed CSS-Eigv obtain highest performance in most cases.

Table 3 presents the AUARC results of sampled generations on the TriviaQA, CoQA, and NQ datasets using LLMs, with the GPT score as the correctness criterion. Responses generated by GPT across all datasets are omitted due to their exceptionally high accuracy (over 95%) – it would be unfair to compare GPT generations with those from LLaMA and OPT. Our method outperforms other baselines, where CSS-Ecc shows more improvements to estimate uncertainty for generations of LLMs.

In summary, the graph Laplacian methods (L-GL and ours)

outperform the white-box methods of semantic entropy and p(true), demonstrating the effectiveness of spectral clustering in analyzing semantic relations. Our proposed method exhibits superior uncertainty estimation compared to L-GL, indicating more effective extraction of semantic relations through the contrastive method.

## 6 ABLATION STUDY

We conducted an extensive ablation study alongside our main experiments to evaluate the necessity of applying dimension reduction to Contrastive Semantic Similarity (CSS) feature maps. We tested CSS feature maps of various dimensions on sampled generations from LLaMA, OPT, and GPT on the CoQA dataset, employing the Eccentricity metric of our proposed method. We utilized the original CSS features with a dimension of 512, and reduced dimensions of 128 and 64 using PCA and UMAP [McInnes et al., 2018] techniques. The results, as shown in Table 4, indicate that a certain level of dimension reduction can enhance AUARC

Table 3: Results of AUARC with GPT score as the correctness criterion, on sampled generation by LLaMA and OPT on dataset of TriviaQA, CoQA and NQ. ACC is accuracy, NumSem is Number of Semantically Distinct Answers (NumSem), and LexiSim means Lexical Similarity. L-GL is Graph Laplacian with NLI Classifier Logits, including EigV, Ecc and Deg sub-methods. WB means white-box methods as semantic entropy (SE) and p(true) require token-level logits access. Our proposed methods include three sub-methods, CSS-EigV, CSS-Ecc, and CSS-Deg, where CSS stands for contrastive semantic similarity. All results are shown in percentages for clarity. Best results are in bold for each dataset.

|  |  | TriviaQA | | CoQA | | NQ | |
|---|---|---|---|---|---|---|---|
|  |  | LLaMA | OPT | LLaMA | OPT | LLaMA | OPT |
|  | Acc | 61.18 | 25.75 | 62.46 | 51.81 | 23.63 | 8.60 |
|  | Oracle | 87.03 | 54.72 | 86.29 | 79.41 | 47.67 | 23.28 |
|  | NumSem | 78.78 | 39.46 | 67.58 | 60.41 | 28.18 | 10.36 |
|  | LexiSim | 80.32 | 45.68 | 78.17 | 71.46 | 40.15 | 15.92 |
| L-GL | EigV | 83.52 | 50.54 | 80.21 | 72.46 | 40.02 | 17.20 |
|  | Ecc | 83.64 | 50.42 | 80.14 | 71.73 | 40.16 | 17.82 |
|  | Deg | 84.61 | 51.06 | 79.34 | 72.51 | 40.81 | 17.43 |
| WB | SE | 79.15 | 51.11 | 78.83 | 70.75 | 36.03 | 17.40 |
|  | P(true) | 64.98 | 20.25 | 64.04 | 50.23 | 24.72 | 7.63 |
| Ours | CSS-EigV | 84.76 | 50.16 | 81.21 | 73.67 | 41.15 | 18.20 |
|  | CSS-Ecc | 84.95 | 51.24 | **82.66** | **73.38** | **42.39** | **18.65** |
|  | CSS-Deg | **86.03** | **52.35** | 81.28 | 72.96 | 41.76 | 18.59 |

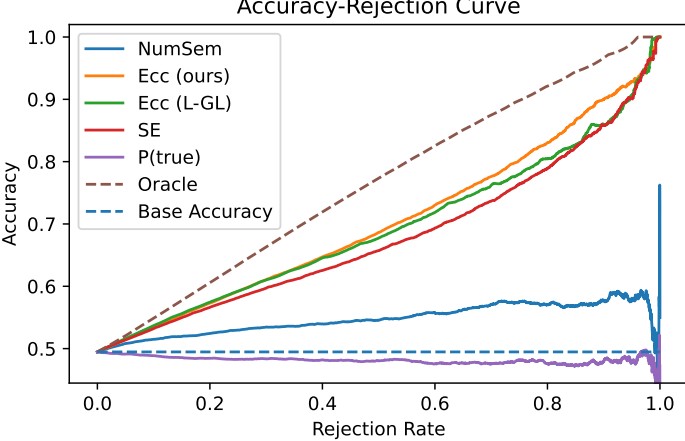

Figure 2: The accuracy-rejection curve for OPT sampled generations for CoQA, with Rouge-L>0.3 as the correctness criterion to obtain the base accuracy. After ranking samples based on their uncertainty scores obtained by the listed methods, we reject samples with higher uncertainty and calculate the accuracy for the remaining data. Oracle represents the highest performance of the model, where the model is perfectly calibrated and all rejected samples are wrongly predicted. We compare the Eccentricity of our method (ECC (ours)) with other baselines, namely p(true), semantic entropy (SE), Eccentricity in L-GL (Ecc (L-GL)), and number of Semantically distinct answers (NumSem).

and AUROC results, suggesting benefits for improved clus-

Table 4: Results of feature reduction on Eccentricity in proposed method (CSS-Ecc) with LLaMA, OPT, and GPT for sampled generations on CoQA Dataset. The original feature dimension is 512, which then reduced to 128 and 64 via PCA and UMAP. Results for our proposed method is underscored.

|  |  | AUARC | | | AUROC | | |
|---|---|---|---|---|---|---|---|
|  | Features | LLaMA | OPT | GPT | LLaMA | OPT | GPT |
| Original | 512 | 80.17 | 71.04 | 84.23 | 77.28 | 74.92 | 77.45 |
| PCA | 128 | 80.54 | 71.12 | 86.89 | 78.25 | 76.75 | 78.52 |
|  | 64 | 80.83 | 71.36 | 87.34 | 78.40 | 76.70 | 80.04 |
| UMAP | 128 | 79.95 | 71.09 | 84.22 | 78.15 | 75.21 | 77.85 |
|  | 64 | 80.52 | 71.16 | 85.64 | 78.46 | 75.67 | 79.58 |

Table 5: Comparison of utilizing NLI logits and NLI feature maps based graph Laplacian on sampled generations of LLaMA and GPT on TriviaQA dataset. The AUARC and AUROC results are based on GPT score for correctness. 'L-GL' denotes the graph Laplacian based on NLI logits, and 'F-GL' represents the graph Laplacian based on NLI feature maps. These results are compared across three sub-methods of uncertainty quantification (UQ): Eigenvalue (EigV), Eccentricity (Ecc), and Degree Metric (Deg).

|  |  | AUARC | | AUROC | |
|---|---|---|---|---|---|
|  |  | LLaMA | OPT | LLaMA | OPT |
| L-GL | EigV | 83.52 | 50.54 | 84.90 | 86.09 |
|  | Ecc | 83.64 | 50.42 | 86.43 | 86.86 |
|  | Deg | 84.61 | 51.06 | 84.21 | 86.60 |
| F-GL | EigV | 83.54 | 50.48 | 84.95 | 85.92 |
|  | Ecc | 83.62 | 51.62 | 86.53 | 86.95 |
|  | Deg | 84.65 | 51.36 | 84.16 | 87.12 |

tering. UMAP's results were slightly inferior to those of PCA; therefore, we used the reduced dimension of 64 by PCA for our main experiments.

Moreover, we argue that NLI classifier logits lack substantial semantic clustering information, as they represent predicted probabilities. To verify this claim, we compared the NLI logits-based graph Laplacian (L-GL) with NLI feature maps extracted from the off-the-shelf DeBERTa model [He et al., 2020] as the basis for the graph Laplacian (F-GL) on sampled generations from LLaMA and GPT on the TriviaQA dataset. The results for EigV, Ecc, and Deg, as presented in Table 5, show that the overall performance of F-GL is marginally better than that of L-GL, indicating that feature maps contain more clustering information than mere probabilities.

CLIP is trained on a contrastive objective using a dataset containing image-caption pairs, where the text encoder is specifically trained on image captions. Despite both using textual data, the domain of image captions can differ significantly from the NLP corpus that language models are trained on. As a result, employing such an image-caption-focused text embedding to evaluate text generated by LLMs

Table 6: Comparing of CLIP text encoder and language models of BERT, DeBERTa and Sentence-BERT for feature embedding with TriviaQA dataset on LLaMA sampled generations. The AUARC and AUROC results are based on GPT score for correctness on evaluation metric Eccentricity (CSS-Ecc). All feature embeddings are without feature reduction, and the best result is in bold.

| Model | AUARC | AUROC |
|---|---|---|
| BERT | 83.78 | 86.24 |
| DeBERTa | 83.62 | 86.53 |
| Sentence-BERT | 83.72 | 87.02 |
| CLIP | **84.32** | **87.19** |

Table 7: Comparing Rouge-L and METEOR as correctness criteria on generated responses by LLaMA on the TriviaQA dataset. The AUARC and AUROC results are on evaluation metric of Eccentricity (Ecc) in L-GL and ours.

| Evaluation Metric | Method | AUARC | AUROC |
|---|---|---|---|
| Rouge-L | L-GL | 80.20 | 83.66 |
| | Ours | 81.29 | 85.17 |
| METEOR | L-GL | 80.32 | 83.79 |
| | Ours | 81.35 | 85.22 |

may raise concerns. We therefore conducted experiments to compare CLIP text encoder and regular Language Models BERT [Devlin et al., 2018], DeBERTa [He et al., 2020] and Sentence-BERT [Reimers and Gurevych, 2019]. We used these language models for feature embedding with TriviaQA dataset on LLaMA sampled generations evaluated with GPT correctness score. Table 6 shows that CLIP outperforms other language models, suggesting CLIP yields more accurate text-text similarity assessments.

Following previous work, we utilized Rouge-L as the correctness measurement for sampled generations of LLMs, to ensure a fair comparison with previous studies [Kuhn et al., 2023, Lin et al., 2023]. However, as the n-gram based metric, Rouge-L may fail to evaluate lexical different but semantically similar sentences, whereas LLM-generated text is more semantically driven. To address this limitation, we employed METEOR, a metric that incorporates more semantic features than simple lexical overlap, as an evaluation criterion for the generated responses of LLMs. Interestingly, results in Table 7 shows that METEOR follow a similar trend with Rouge-L in AUROC and AUARC.

## 7 DISCUSSION

The empirical evidence shown in tables 1, 2 and 3 from extensive experiments across multiple datasets and evaluation metrics firmly establishes the superiority of our proposed contrastive semantic similarity over existing methods, especially the most recent NLI logits-based approaches. Our

proposed CLIP-based semantic similarity further learns contrastive features between text pairs, demonstrating better semantic clustering compared to the baselines.

Our extensive ablation study and novel findings demonstrate the superiority of our intuitive and simple, yet effective, approach. The study presented in Table 4 demonstrates the benefits of dimension reduction on feature maps for improved clustering with techniques of PCA and UMAP. Additionally, NLI logits represent the predicted probabilities for the labels "entailment," "contradiction," and "neutral." However, they lack comprehensive latent semantic features, as logits are primarily trained to identify labels. Semantic relationships between text pairs can be implicit, making a feature map a better semantic representation than NLI logits. Our ablation study, shown in Table 5, compares NLI logits and the NLI feature map from DeBERTa, revealing that feature maps contain more potential semantic clustering information than logits.

Furthermore, we investigate the effectiveness of text feature extraction between CLIP and regular language models, with results shown in Table 6. CLIP's ability to extract contrastive features from input pairs has made it widely applicable for understanding alignments between image-text pairs [Radford et al., 2021, Ramesh et al., 2022, Hou et al., 2022] and as well as between image-image pairs [Yu et al., 2024]. While previous studies have focused on CLIP's use with image-text and image-image data, we extend its application to investigate image-free contrastive semantic feature extraction, thus broadening the scope for which CLIP can be utilized.

In addition, to address the limitations of the n-gram-based metric ROUGE-L, we incorporate METEOR as an evaluation criterion for the generated responses to better capture semantic similarities. Although both metrics reveal similar trends in their results shown in Table 7, the use of METEOR provides a more accurate and meaningful evaluation than ROUGE-L. We will expand our experiments to include additional metrics that more effectively account for semantic features, thereby providing a more accurate evaluation of generated text of LLMs.

Our research also contributes to the ongoing discourse on the trustworthiness of large language models, offering a pragmatic solution to the challenge of selective answering in question-answering systems. The demonstrated efficacy of our CSS (shown in Figure 2) in identifying and rejecting unreliable generations holds significant implications for the development of more trusted LLMs applications.

In terms of bias and fairness, vision-language models invariably exhibit varying degrees of bias, as highlighted in the work [Radford et al., 2021] using the FairFace dataset, which includes race, gender, and age subgroups. However, our current datasets lack subgroup information, and our primary objective is to develop an effective UQ method for

LLMs. In future research, we will address bias and fairness issues by incorporating relevant subgroup data into our uncertainty estimation processes, ensuring a comprehensive evaluation and improvement of fairness in our models.

# 8 CONCLUSION

In this paper, we proposed a novel UQ technique for LLMs using Contrastive Semantic Similarity (CSS) to capture insightful semantic relationships between text pairs. By adapting the CLIP text encoder and utilizing spectral clustering, our method accurately estimates the uncertainty of LLM-generated responses than SOTA techniques.

Our extensive experiments demonstrated that our approach outperforms existing UQ methods, revealing richer semantic information than NLI logits. We also showed the superiority of contrastive feature extraction of CLIP over regular language models, expanding its application scope in language generation. Furthermore, our exploration of the METEOR metric provided a more comprehensive assessment of semantic relationships compared to ROUGE-L, enhancing evaluation criteria for generated texts. Our method also improves selective NLG by more effectively identifying unreliable responses. Future work will focus on further confidence and uncertainty calibration techniques and exploring the application of our method to a broader range of NLG tasks.

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
