# OpenReview forum: "CSS: Contrastive Semantic Similarities for Uncertainty Quantification of LLMs"
_auai.org/UAI/2024/Conference — UAI 2024 poster_

### Official Review · Reviewer_iKBb · 2024-03-08

**Q2-1 Originality-Novelty:** 2
**Q2-2 Correctness-Technical Quality:** 3
**Q2-5 Clarity Of Writing:** 2

**Q1 Summary And Contributions:**

The paper proposes a method to quantify the uncertainty in the generations of an LLM. The motivation is that such a method could be used in a selective NLG setting (such as selective question answering) to selectively not answer questions if the uncertainty is high (as the answer is more likely to be wrong in such cases). The proposed method is built on prior work that uses spectral clustering of LLM generations; the number of clusters is a measure of the uncertainty. The paper proposes to modify the similarity metric used for clustering to one based on the CLIP text encoder instead of using NLI logits as in the prior work, arguing that it captures better the semantic relation between texts. The method is evaluated for the selective question answering task on a variety of datasets using three different LLMs, namely LLaMA, OPT and GPT, and it is compared with several baselines from prior work and with an Oracle. The results in terms of AUARC and AUROC show that quantifying the uncertainty, and selectively not answering when the uncertainty is high, improves the accuracy, and that the proposed method for uncertainty quantification outperforms the baselines.

**Q2-3 Extent To Which Claims Are Supported By Evidence:**

3: Good: the main claims are supported by convincing evidence (in the form of adequate experimental evaluation, proofs, (pseudo-)code, references, assumptions).

**Q2-4 Reproducibility:**

2: Fair: key resources (e.g. proofs, code, data) are unavailable but key details (e.g. proof sketches, experimental setup) are sufficiently well-described for an expert to confidently reproduce the main results.

**Q3 Main Strengths:**

1. The method proposed in the paper outperforms the baseline from prior work. It indeed suggests that the CLIP encoder based similarity is better for uncertainty quantification using spectral clustering, which is a novel and useful finding.
2. The experimental evaluation is thorough: using multiple LLMs, several datasets, and a variety of baselines from prior work.

**Q4 Main Weakness:**

1. The argument for using CLIP encoder based similarities instead of NLI logits is not well made. It is not clear why taking the Hadamard product of CLIP encoder representations should be expected superior to NLI logits in capturing semantic relations. Also, would taking the product between representations from other text embedding approaches be superior as well? Why or why not?
2. Novelty is fair at best. The paper largely follows prior work in the problems addressed and methods used (spectral clustering, with the same uncertainty measures), with only a small change in terms of the similarity metric used.
3. As defined, css(r_i, r_j) is a vector, being the Hadamard product of the representations. It is not clear how it is then used to construct the adjacency matrix. Are the entries summed?
4. Clarity of writing can be improved in general. There are several places where mistakes are present in equations (see equations (2), (5), degree equation).

**Q5 Detailed Comments To The Authors:**

1. A more detailed and thorough explanation may be provided for why CLIP based similarities may be expected to outperform NLI logits (the motivation for this experiment).
2. As defined, css(r_i, r_j) is a vector, being the Hadamard product of the representations. It is not clear how it is then used to construct the adjacency matrix. Are the entries summed?
3. The paper may be subjected to a thorough proof reading to fix mistakes in equations (see equations (2), (5), degree equation) , and grammar and spelling errors.

**Q9 Complying With Reviewing Instructions:**

Yes

---

> ### Author Rebuttal · Authors · 2024-04-07
>
> We would like to thank the reviewer for the comment and recognising our proposed method of uncertainty quantification for LLMs as ‘ a novel and useful finding’ and ‘thorough evaluation’. We have addressed all concerns raised by the reviewer as below.
>
> R1:NLI logistics is the predicted probability of labels entailment, contradiction and neutral. It hardly contains the latent semantic features as logits are trained to learn labels. However, semantic relations between text-pairs can be implicit. Hence, a feature map is a better semantic representation than NLI logits. Our ablation study in Table 5 compares the NLI logits and NLI feature map by DeBERTa, which shows that feature maps are shown to contain more potential semantic clustering information than logits. Our proposed CLIP-based semantic similarity further learns the contrastive feature between text pairs, which shows better semantic clustering than baselines. We will add this detailed explanation in the final version. We have also conducted additional experiments to compare the performance of CLIP encoder and language models of BERT and RoBERTa (reviewer wEJ5, Q4.2).
>
>
> R2: Apology for the confusion. The adjacency matrix W is formed with the symmetrisation of w_{i,j}, where w_{i,j} is the CLIP contrastive feature of (r_i, r_j). We will add this information in the final version.
>
>
> R3: Thank you for the suggestion. We have conducted proofreading and addressed issues including: (i) typos in equations (2), (5), and the degree equation (reviewer WyVN, comment R5); (ii) citation inconsistencies and grammatical errors (reviewer WyVN, comments R2, R4); (iii) unclear explanations and repetition in the introduction and related work sections, as well as minor typos (reviewer WyVN, comments R1, R3). All modifications will be included in the final version.

---

### Official Review · Reviewer_wEJ5 · 2024-03-22

**Q2-1 Originality-Novelty:** 1
**Q2-2 Correctness-Technical Quality:** 2
**Q2-5 Clarity Of Writing:** 3

**Q1 Summary And Contributions:**

This paper proposes a method to quantify uncertainty in LLM generated responses. Specifically, given text pairs generated by LLM for the same query, output a uncertainty score for the LLM.  The paper is based on (Lin et al. 2023). Specifically, (Lin et al. 2023) proposed to use NLI logits to measure the similarity of two texts in the pairs, then compute Graph Laplacian of the similarity vector to compute the uncertainty score.

This paper’s novelty is to use another text pair similarity metrics. The authors replace the NLI-logits used in (Lin et al. 2023) with the dot product of the output of the publicly available pretrained CLIP text encoder.

**Q2-3 Extent To Which Claims Are Supported By Evidence:**

2: Fair: the main claims are somewhat supported by evidence (but the experimental evaluation may be weak, or does not match entirely with the claims, important baselines may be missing, proofs contain important ideas but lack rigor, algorithmic details are only discussed superficially, references are imprecise, assumptions are not sufficiently motivated or explicated, etc.).

**Q2-4 Reproducibility:**

3: Good: key resources (e.g. proofs, code, data) are available and key details (e.g. proofs, experimental setup) are sufficiently well-described for competent researchers to confidently reproduce the main results.

**Q3 Main Strengths:**

1. The usage of CLIP text encoder to compute text pair similarity and using it in (Lin et al. 2023)’s framework appears to be new.
2. The experiment shows superior performance over related methods.

**Q4 Main Weakness:**

1. (major) Limited novelty. The paper appears to be based on the problem formulation and framework proposed in (Lin et al. [2023]). The only change is to replace the text pair similarity measure with the dot product of the output of the publicly available pretrained CLIP text encoder.
2. (major) The usage of the CLIP text encoder is questionable. CLIP is trained on the contrastive objective using a dataset of image and caption pairs. Therefore, the text encoder is likely trained more heavily on the domain of image captions, likely very different from the NLP corpus that LLMs are trained on. Using such image-caption-focused text embedding to evaluate LLM-generated text is questionable. Why is a CLIP text encoder preferred over a regular Language Model (such as BERT, RoBERTa, T5, or GPT)?
3. (major) Didn’t compare with a Language Model (LM) baseline. One straight-forward baseline is to replace the text pair similarity measure in (Lin et al. [2023]) with a LM-based model. I.e., use a regular Language Model (such as BERT, RoBERTa, T5, or GPT) to obtain the embedding of the text pairs, then use the cosine difference as the similarity measure. This could reveal the performance of using the CLIP text encoder.
4. (medium) The evaluation metrics are Rouge-L and “GPT correctness score”. Rouge-L appears to be inappropriate, because it is a n-gram based metric, while LLM-generated texts are clearly semantic-based. ROUGE-L will fail to evaluate lexical different but semantically similar sentences.
5. (minor) “GPT correctness score” is not defined. I had to check the reference (Lin et al. 2023) to figure it out. The “ACC” in the result tables is not defined.

**Q5 Detailed Comments To The Authors:**

My comments are in the weakness section.

**Q9 Complying With Reviewing Instructions:**

Yes

---

> ### Author Rebuttal · Authors · 2024-04-07
>
> We would like to thank the reviewer for the comment and recognising our proposed method of uncertainty quantification for LLMs as ‘new approach ’ and ‘superior performance over related methods’. We have addressed all concerns raised by the reviewer as below.
>
> R1: We design an innovative way to extract text-text similarity features using CLIP and dot product, where CLIP is originally trained for image-text similarity. Our CLIP-extracted contrastive features contain more semantic relations than NLI logits, which is simple yet effective. Our extensive evaluation reflects the effectiveness of our approach, with averagely 1.5 - 2.5% performance improvement. Furthermore, to address the comment Q4.4, we have also investigated the use of METEOR over Rouge-L to obtain the correctness of generated responses from LLMs with better semantic evaluation. This investigation will be added as the ablation study in the final version.
>
>
> R2: Thanks for the suggestion. We have conducted experiments to compare CLIP text encoder and regular Language Models. We present the result of using BERT for feature embedding, with TriviaQA dataset on LLaMA sampled generations evaluated with GPT correctness score (threshold is 0.7). The evaluation metric is Eccentricity (CSS-Ecc) without feature reduction. From the table below, CLIP outperforms BERT and RoBERTa in terms of AUARC and AUROC.
>
> | Model  | AUARC | AUROC |
> | ----- | ----- |----- |
> | BERT  | 83.78 | 86.24 |
> | DeBERTa  | 83.62 | 86.53 |
> | CLIP | 84.32 | 87.19 |
>
> R3: From the R2, it is evident that CLIP surpasses BERT and RoBERTa in terms of AUROC and AUPRC, demonstrating superior uncertainty estimation. This finding also implies that CLIP yields more accurate text-text similarity assessments. However, utilising cosine similarity as a metric might be misleading given our focus on UQ in LLMs. Furthermore, comparing the embedding performance of CLIP with BERT and RoBERTa through cosine similarity on generated text pairs becomes problematic due to the absence of a ground truth for the similarity between the generated content and the reference answer.
>
>
> R4: We appreciate the reviewer's comment regarding Rouge-L's limitations in evaluating semantic similarity. We agree and have incorporated METEOR, a metric that considers more semantic features than just lexical overlap. We conducted experiments using METEOR as the evaluation criterion on LLaMA's generated responses from the TriviaQA dataset, focusing on the Ecc metric (Eccentricity), by comparing our method and the L-GL (Lin et.al, 2023 [1]). Interestingly, the results with METEOR follow a similar trend to those obtained with Rouge-L (presented in the table below). We will include the METEOR results in the final ablation study for a more comprehensive evaluation.
> However, to maintain consistency with recent works [1, 2] that employed Rouge-L, we will retain the original table using Rouge-L in the main paper for a fair comparison.
>
>
> | Evaluation Metrics  | Method | AUARC | AUROC|
> | ----- | ----- |----- | ----- |
> | Rouge-L  | L-GL | 80.20 |83.66 |
> | Rouge-L  |Ours | 81.29 |85.17 |
> | METEOR | L-GL | 80.32 |83.79 |
> | METEOR | Ours | 81.35|85.22 |
>
> R5: GPT correctness score is provided by gpt-3.5-turbo from the OpenAI API, which assigns a correctness score between 0 and 1 for the similarity between given reference answer and generated responses. ACC refers to accuracy in table 2. We will add these explanations in the final version.
>
> [1] Lin, Z., Trivedi, S. and Sun, J., 2023. Generating with confidence: Uncertainty quantification for black-box large language models. arXiv preprint arXiv:2305.19187
>
> [2] Kuhn, L., Gal, Y. and Farquhar, S., 2022, September. Semantic Uncertainty: Linguistic Invariances for Uncertainty Estimation in Natural Language Generation. In The Eleventh International Conference on Learning Representations.

---

### Official Review · Reviewer_xZiE · 2024-03-22

**Q2-1 Originality-Novelty:** 2
**Q2-2 Correctness-Technical Quality:** 3
**Q2-5 Clarity Of Writing:** 3

**Q1 Summary And Contributions:**

This work proposes a new method for uncertainty quantification for language models, based on similarity metrics over model generations. It builds off of a recent work that proposes a metric using NLI predictions to cluster together sentences with similar meanings; it then estimates uncertainty as the entropy over cluster membership. The authors propose two main changes to this previous procedure (1) to instead use differences between text embeddings (learned via the CLIP algorithm) in order to quantify text similarities and (2) to use spectral clustering rather than k-means clustering. A nice feature of this method is that it does not require logits from the model that's under inspection. They perform experiments on the ability of this metric to determine when to abstain from providing an answer. More specifically, they use the AUARC metric with question-answering datasets. They find that their method consistently improves on this criterion across several datasets.

**Q2-3 Extent To Which Claims Are Supported By Evidence:**

3: Good: the main claims are supported by convincing evidence (in the form of adequate experimental evaluation, proofs, (pseudo-)code, references, assumptions).

**Q2-4 Reproducibility:**

3: Good: key resources (e.g. proofs, code, data) are available and key details (e.g. proofs, experimental setup) are sufficiently well-described for competent researchers to confidently reproduce the main results.

**Q3 Main Strengths:**

* The paper weighs in on a very relevant topic: uncertainty for natural language generation
* The authors compare against a good number of baseline methods and across several datasets in their experiments
* The proposed adjustments to the uncertainty quantification metric of Kuhn et. al. 2023 are intuitive and seem like they would make the metric more widely applicable.

**Q4 Main Weakness:**

* There are aspects of the work that are unclear: what are the details of the model used to produce the CLIP embeddings? Is it a pretrained model or are the authors training the model themselves?
* The justification for using CLIP embeddings is somewhat weak: why should we believe that these embeddings in particular encode semantic features of text? Is there evidence supporting this assumption?
* The contributions of the work are limited; the use of spectral clustering for determining the number of semantic sets has already been proposed (Lin et. al 2023)
* No standard deviation or significance tests are performed, so its impossible to tell the statistical significance of the results

**Q5 Detailed Comments To The Authors:**

Please see questions in "weaknesses" section above

**Q9 Complying With Reviewing Instructions:**

Yes

---

> ### Author Rebuttal · Authors · 2024-04-07
>
> We would like to thank the reviewer for the comment and recognising our proposed method of uncertainty quantification for LLMs as ‘very relevant topic ’, ‘good number of baseline methods’, ‘extensive datasets and experiments’, and ‘widely applicable’. We have addressed all concerns raised by the reviewer as below.
>
> R1: We use the pre-trained CLIP model by using Huggingface library, which is trained on a dataset of about 400 million image-text pairs collected from the Internet. The information will be added in the section ‘Experiments’.
> From Hugging Face library, the sample code is below:
>
> from transformers import CLIPProcessor, CLIPModel \
> model = CLIPModel.from_pretrained("openai/clip-vit-base-patch32")
>
> R2:
> We are using CLIP for text feature extraction instead of text-image cosine similarity, for which the CLIP was trained originally. To validate the effectiveness of CLIP, we have done additional experiments comparing text feature extraction of CLIP over other language models such as BERT and RoBERTa to address the comment of the reviewer (wEJ5, comment Q4.2). Moreover, with the capability of extracting contrastive features of input pairs, CLIP has been used in various applications to understand the alignments between image-text and image-image [1, 2, 3, 5]. The current study demonstrates that the semantic relation between text pairs lies more in the implicit context than the token overlapping [4]. Hence, our work explores the capability of contrastive semantic feature extraction of CLIP for text-text pairs and demonstrates improvement over baselines.
>
> R3: Our main contribution lies in extracting contrastive semantic features from text instead of spectral clustering. We design an innovative way to extract text-text similarity features using CLIP and dot product, where CLIP is originally trained for image-text similarity. Our CLIP-extracted contrastive features contain more semantic relations than NLI logits, which is simple yet effective. Our extensive evaluation reflects the effectiveness of our approach. Furthermore, to address the reviewer (wEJ5, comment Q4.4), we have also investigated the use of METEOR over Rouge-L to obtain the correctness of generated responses from LLMs with better semantic evaluation. This investigation will be added as the ablation study in the final version.
>
> R4: Thanks for the valuable suggestion. We have calculated the standard deviation (std) for all metrics. Our std is smaller than most of the baselines, which reflect the consistency of our method. Due to the space limitation, we hereby provide the AUARC result of TriviaQA with sampled generations from LLaMA, with GPT correctness score (>0.7) as the criteria. We will update all results in the final version.
>
>
> | Method  | Sub-method | AUARC |
> | ----- | ----- |----- |
> | NumSem  | -- | 78.78±0.17 |
> | LexiSem  | -- | 80.32±0.05 |
> | L-GL  | EigV | 83.52±0.08 |
> | L-GL  | Ecc | 83.64±0.06 |
> | L-GL  | Deg | 84.61±0.06 |
> | WB  | SE | 79.15±0.08 |
> | WB  | p(true)| 64.98±0.11 |
> | Ours  | CSS-EigV | 84.76±0.06 |
> | Ours  | CSS-Ecc | 84.95±0.04 |
> | Ours  | CSS-Deg | 86.03±0.05 |
>
>
> [1] Li, M., Xu, R., Wang, S., Zhou, L., Lin, X., Zhu, C., Zeng, M., Ji, H. and Chang, S.F., 2022. Clip-event: Connecting text and images with event structures. In Proceedings of the IEEE/CVF conference on computer vision and pattern recognition (pp. 16420-16429).
>
> [2] Ramesh, A., Dhariwal, P., Nichol, A., Chu, C. and Chen, M., 2022. Hierarchical text-conditional image generation with clip latents. arXiv preprint arXiv:2204.06125, 1(2), p.3.
>
> [3] Liu, H., Li, C., Wu, Q. and Lee, Y.J., 2024. Visual instruction tuning. Advances in neural information processing systems, 36.
>
> [4] Kuhn, L., Gal, Y. and Farquhar, S., 2022, September. Semantic Uncertainty: Linguistic Invariances for Uncertainty Estimation in Natural Language Generation. In The Eleventh International Conference on Learning Representations.
>
> [5] Yu, C., Liu, X., Wang, Y., Zhang, P. and Lu, H., 2024, March. TF-CLIP: Learning Text-Free CLIP for Video-Based Person Re-identification. In Proceedings of the AAAI Conference on Artificial Intelligence (Vol. 38, No. 7, pp. 6764-6772).

---

### Official Review · Reviewer_WyVN · 2024-03-22

**Q2-1 Originality-Novelty:** 2
**Q2-2 Correctness-Technical Quality:** 3
**Q2-5 Clarity Of Writing:** 2

**Q10 Ethical Concerns:**

No.

**Q1 Summary And Contributions:**

This paper considers the problem of uncertainty quantification for LLMs for QA, i.e. estimating the confidence of the generated answer for a question. The setting in the paper is one where the LLM is treated as a black-box, similar to prior work such as Kuhn et al. and Lin et al. The main contribution is the proposal of a similarity metric for pairs of text, which is then leveraged to gauge consistency with respect to multiple generated samples from the LLM based on the prior approaches. Several experiments are conducted to validate the proposed similarity metric.

**Q2-3 Extent To Which Claims Are Supported By Evidence:**

3: Good: the main claims are supported by convincing evidence (in the form of adequate experimental evaluation, proofs, (pseudo-)code, references, assumptions).

**Q2-4 Reproducibility:**

3: Good: key resources (e.g. proofs, code, data) are available and key details (e.g. proofs, experimental setup) are sufficiently well-described for competent researchers to confidently reproduce the main results.

**Q3 Main Strengths:**

The main strength of the paper is that it is a fairly extensive empirical study as it compares with several relevant baselines for a few QA tasks. I note that around half of the paper’s content consists of the description and explanation of experimental results. From that perspective, I think the empirical nature of the paper is substantial.

**Q4 Main Weakness:**

My main concern with the paper is that the methodological contribution in the work appears to be unsubstantial and not completely justified. The connection to CLIP was unclear to me. CLIP is about the multi model setting and identifying a joint embedding. What is the contrastive aspect here when the pairs are both textual? Why not just use regular text embeddings for similarity? Note that more than half of Section 3 is about prior work. The extension beyond the work of Lin et al. comes across as minor.

Another major weakness of the paper is that there is imprecision about prior work which could make the paper hard to read, particularly for those who are not familiar with the work of Kuhn et al. and Lin et al. I will describe some examples in my detailed comments.

Overall, I feel like the paper doesn’t do enough to justify its proposed method and comes across as quite ad-hoc. If the authors can do better in this regard, I think the paper would be stronger. I’m willing to reconsider my assessment during the discussion phase.

**Q5 Detailed Comments To The Authors:**

Here are some additional questions, comments, and suggestions:

In the second paragraph of the paper, I suggest the authors make it clear that verbal methods such as from Kadavath et al. are also black-box methods. I think this is unclear from the last sentence in this paragraph.

Should it be “logits maps” or “logit maps” on page 1?

There seems to be some repetition of related work in Sections 1 and 2. I don’t think the authors used the space wisely. More space should be allocated to justifying the proposed similarity approach, possibly with some examples.

There are several issues with citation styles in the paper; see the first paragraph of Section 2, for example.

There is imprecision in several places throughout the paper, which makes it hard to understand. An example is on page 2 where the authors write
“uncertainty is estimated as differences among clusters”. Section 3 is the most important section in the paper, yet it is the most unclear. What is C in the section on semantic entropy? Description about D_{ij} seems imprecise because D is a diagonal matrix. What is the justification of doing one minus lambda in equation (2)? And should this be lambda_k in the equation instead of lambda_n? A lot of the content here seems like unclear descriptions of the prior work. I noted several typos on page 3, which I hope the authors will fix.

**Q9 Complying With Reviewing Instructions:**

Yes

---

> ### Author Rebuttal · Authors · 2024-04-07
>
> We would like to thank the reviewer for the comment and recognising our proposed method of uncertainty quantification for LLMs as ‘extensive empirical study ’, ‘proposal of a similarity metric for pairs of text’, and ‘substantial experiments’. We have addressed all concerns raised by the reviewer as below.
>
> R1: Apology for the confusion. A white-box method is one that leverages internal model information, such as token-level predicted probabilities  (often called logits or confidence scores)  from LLMs. Methods proposed by Kadavath et al. (2022) and Lin et al. (2022b) are white-box methods as they ask the model itself about the confidence of its prediction. We have re-written this part of the introduction by summarising the second and third paragraph, to address this comment and the third comment of repetition of related work in Sections 1 and 2. Please find the revised text as below:
>
> The study of UQ in LLMs has gained significant attention recently. Most existing methods are white-box, relying on either calculating entropy from predicted probabilities [Malinin and Gales, 2020, Kuhn et al., 2023] or querying models for their prediction confidence [Lin et al., 2022b, Kadavath et al., 2022]. However, these techniques often require task-specific labels, additional training data, or white-box access to the internal model information. Black-box UQ strategies address this by analysing the consistency of information across model generations. Techniques like n-gram overlap [Fomicheva et al., 2020] assess surface-level similarity, while more recent approaches explore semantic equivalence [Kuhn et al., 2023, Lin et al., 2023]. These methods cluster sentences based on meaning to estimate uncertainty, with a higher number of clusters indicating greater semantic diversity and thus higher LLM uncertainty. However, a key limitation lies in using Natural Language Inference (NLI) classifier logits to measure semantic equivalence. Logits represent class probabilities, not the semantic features needed for accurate clustering. This highlights the need for more sophisticated features that better capture the true semantic relationships between generated texts.
>
> R2: Apology for the typo. It is just “logits” by following papers [1,2]. We have rectified it in the entire paper and will update in the final version.
>
>
> R3: We have summarised the second and third paragraph in the introduction section to address the repetition issue. Please refer to R1 for the detailed revised version. Now we have additional space to add ablation studies to address comments of reviewer (wEJ5, comment Q4.2, 4.4).
>
> R4: Thanks for pointing out this inconsistency. We have fixed issues of citations at the end of sentences but without the square bracket, also citations at the beginning of sentences. We have fixed all other inconsistencies and will be reflected in the final version.
>
> For example, we fix the citation at the first paragraph of Section 2 by rewriting the sentence as below:
> Moreover, a recent study introduces a novel entropy-based uncertainty measure called semantic entropy, incorporating linguistic invariances created by shared meanings [Kuhn et al., 2023].
>
>
> R5(a). Apologies for the confusion. We have fixed the imprecision which is in the R2. More specifically, the rectified similar sentence is: These methods cluster sentences based on meaning to estimate uncertainty, with a higher number of clusters indicating greater semantic diversity and thus higher LLM uncertainty.
>
> R5(b): Thanks for pointing it out. C is the number of semantic clusters, where semantically equivalent sentences fall into the same cluster. The explanation will be added in section 3 of the final version.
>
> R5(c): Thanks for pointing out the description of D_{ij}. It is rectified as below:
> D_{ij} is non-zero when i=j and r_i and r_j are semantically equivalent. Otherwise D_{ij} is zero.
>
> R5(d): We will add the justification in the final version as below:
> In equation (2), lambda is the eigenvalue in graph Laplacian, which is typically used to determine the number of clusters. Following the works [1,3], Eigenvalues larger than 1 are ignored as only the smallest few eigenvalues carry important information about the clusters. Hence equation (2) picks the max value between 0 and ‘one minus lambda’ to ignore eigenvalues larger than 1.
>
>
> R5(e): Apology for the typo. It should be lambda_k.
>
> R5(f): Thanks for pointing it out. We have fixed the inconsistency of citation style, missing punctuations, and typo in equation (2) and degree matrix. We will update it in the final version.
>
> [1]  Lin, Z., Trivedi, S. and Sun, J., 2023. Generating with confidence: Uncertainty quantification for black-box large language models. arXiv preprint arXiv:2305.19187
>
> [2] Kuhn, L., Gal, Y. and Farquhar, S., 2022, September. Semantic Uncertainty: Linguistic Invariances for UQ in NLG. The 11th ICLR.
>
> [3] Von Luxburg, U., 2007. A tutorial on spectral clustering. Statistics and computing, 17, pp.395-416.

---

### Official Review · Reviewer_Qtvv · 2024-03-24

**Q2-1 Originality-Novelty:** 3
**Q2-2 Correctness-Technical Quality:** 3
**Q2-5 Clarity Of Writing:** 3

**Q1 Summary And Contributions:**

- The paper proposes Contrastive Semantic Similarity with graph Laplacian (C-GL), a novel approach for enhancing uncertainty quantification in natural language generation by utilizing CLIP's feature extraction capabilities.
- C-GL method outperforms existing approaches, including NLI logits-based methods and lexical similarity measures, in estimating the reliability of large language model generations.
- Extensive experiments and ablation studies across multiple datasets and evaluation metrics establish the superiority of C-GL, particularly in its ability to provide a nuanced assessment of semantic similarity and dispersion.
- The research contributes to the field by offering a method that improves the trustworthiness of question-answering systems, addressing the critical need for reliable uncertainty measures in LLM applications.

**Q2-3 Extent To Which Claims Are Supported By Evidence:**

3: Good: the main claims are supported by convincing evidence (in the form of adequate experimental evaluation, proofs, (pseudo-)code, references, assumptions).

**Q2-4 Reproducibility:**

2: Fair: key resources (e.g. proofs, code, data) are unavailable but key details (e.g. proof sketches, experimental setup) are sufficiently well-described for an expert to confidently reproduce the main results.

**Q3 Main Strengths:**

- The paper introduces Contrastive Semantic Similarity with graph Laplacian (C-GL), a novel method for uncertainty quantification in natural language generation, which outperforms state-of-the-art techniques by leveraging the advanced feature extraction capabilities of CLIP.
- It contributes to the trustworthiness of large language models by providing a pragmatic solution for selective answering in question-answering systems, which detects and rejects unreliable generations.
- The research demonstrates through extensive experiments that feature maps contain more potential semantic clustering information than logits, validating the use of feature maps with dimension reduction for better clustering of LLM generations.
- The study also addresses the challenge of measuring uncertainty without requiring task-specific labels, additional training, or fine-tuning, which is beneficial for end-users with limited resources.

**Q4 Main Weakness:**

- The paper's proposed method may not generalize well across all types of LLMs or datasets, as the experiments were conducted on specific benchmark question-answering datasets.
- There is a lack of discussion on the computational resources required for the C-GL method, which could be a limitation for end-users with restricted computational capabilities.
- The reliance on CLIP's feature extraction may introduce biases inherent in the pre-training data of CLIP, which is not addressed in the paper.
- The paper does not provide a clear comparison with all state-of-the-art UQ techniques, as some methods could not be evaluated due to the unavailability of token-level probabilities from the ChatGPT API.

**Q5 Detailed Comments To The Authors:**

**Methodology Generalization**
- Consider testing the C-GL method across a wider range of LLMs and datasets to ensure the robustness and generalizability of the proposed approach.

**Computational Efficiency**
- Provide an analysis of the computational resources required for the C-GL method, as this information is crucial for practical applications and scalability.

**Bias and Fairness**
- Investigate potential biases introduced by CLIP's pre-training data and explore mitigation strategies to ensure fairness in the C-GL method's uncertainty quantification.

**Comparison with State-of-the-Art**
- Expand the comparison to include a broader range of state-of-the-art UQ techniques, especially those that could not be evaluated due to API limitations, to provide a more comprehensive benchmarking.

**Limitations and Future Work**
- Discuss the limitations of the current study and outline potential areas for future research to encourage ongoing improvements in the field of uncertainty quantification for LLMs.

**Q9 Complying With Reviewing Instructions:**

Yes

---

> ### Author Rebuttal · Authors · 2024-04-07
>
> We would like to thank the reviewer for the comment and recognising our proposed method of uncertainty quantification for LLMs as ‘novel method’, ‘outperforms state-of-the-art techniques’, ‘contributes to the trustworthiness’, and ‘extensive experiments’. We have addressed all concerns raised by the reviewer as below.
>
>
> R1: Methodology Generalization:
> In this paper, our work scope is the question-answering task. We utilise three recent SOTA LLMs (before the UAI submission deadline) and three benchmark QA datasets to validate our proposed CSS method. This results in a total of nine evaluations (three LLMs x three datasets) to validate the proposed method. In addition, each question has 20 sampled generations that include diverse and potentially wrong responses to have a more accurate uncertainty estimation for LLMs. Apart from this, we have conducted additional validation via using METEOR, which is a better semantic similarity measurement following the comment of reviewer wEJ5 (Q4, comment 4). All these validations demonstrate the superiority of our method, which we believe is strong evidence of the generalisation of our method.
> Moreover, the previous work [1] only use one LLM architecture (OPT) and the above three dataset to validate their proposed uncertainty quantification (UQ) model; and Lin et al. (2023) [2] uses similar number of architectures and datasets to demonstrate the generalisation of their proposed UQ method.
>
> R2: Computational Efficiency:
> We have calculated the computational cost for our CSS method over the previous work [2]. Our CSS takes about 2.3 seconds to calculate the UQ for a text pair, where previous work [2] takes about 1.2 seconds. This demonstrates a minor computational additional resource for our method, which is still quite fast. For all our experiments, we use the 2 GPUs of Nvidia Tesla P40 with 23 GB RAM. Generating 20 responses for each question takes about 30 - 50 seconds. We will add this information in the section ‘Results’ in the final version.
>
> R3: Bias and Fairness:
> Thanks for the comment. We believe there are always certain levels of bias and fairness issues in all vision-language models, which is also reported in the work [3] with the FairFace dataset (with subgroups of race, gender, and age classification). However, our datasets do not contain sub-group information, and our primary goal is to develop an effective UQ method for LLMs. In future work, we will incorporate bias and fairness in estimating uncertainty in proper datasets. We will add this information to the  ‘Discussion’ section of future work.
>
> R4: Comparison with State-of-the-Art:
> Apologies for the confusion. We utilise the Uncertainty Quantification (UQ) techniques of Semantic Entropy (SE) and p(true), which depend on accessing model-specific information, such as logits or confidence scores. These techniques could not be assessed due to limitations with the API. Our proposed CSS still outperforms SOTA UQ techniques that utilise LLMs as white-box or black-box.
>
> R5: Limitations and Future Work:
> We plan to explore additional tasks of Large Language Models (LLMs) such as text summarization and visual question answering. Furthermore, we will investigate strategies to address the bias and fairness issues in LLMs. A detailed explanation will be included in the 'Conclusion' section of the final version.
>
> [1]  Lin, Z., Trivedi, S. and Sun, J., 2023. Generating with confidence: Uncertainty quantification for black-box large language models. arXiv preprint arXiv:2305.19187
>
> [2] Kuhn, L., Gal, Y. and Farquhar, S., 2022, September. Semantic Uncertainty: Linguistic Invariances for Uncertainty Estimation in Natural Language Generation. In The Eleventh International Conference on Learning Representations.
>
> [3] Radford, A., Kim, J.W., Hallacy, C., Ramesh, A., Goh, G., Agarwal, S., Sastry, G., Askell, A., Mishkin, P., Clark, J. and Krueger, G., 2021, July. Learning transferable visual models from natural language supervision. In International conference on machine learning (pp. 8748-8763). PMLR.

---

### Meta-Review · Area_Chair_iVPA · 2024-04-19

The paper presents an uncertainty quantification technique used to abstain from answering (aka, selective generation) when using LLMs for question answering problems. The technique builds upon 'semantic entropy', prior work where the distribution over responses is re-expressed as a distribution over semantic clusters, these are in turn defined by a model that assigns probability to certain logical entailment relations. The key contribution is to change this clustering algorithm. The paper contributes an approach based on CLIP, the result seems widely applicable. The paper has a strong empirical section, showing that their take on semantic entropy improves on the original formulation.

This paper gathered a lot of feedback, and it's almost a consensus that it is shy of technical novelty. I normally don't mind that too much, trying to weigh the relevance of the findings in relation to the questions that are asked, and the perceived importance of those questions at one point in time. Here's where I struggle (the one question that seemed pretty relevant to me and 4 other reviewers was consistently avoided in the rebuttal).

From the rebuttal, it really looks like the authors insist that the core of the contribution is to use their CLIP-based technique (as opposed to any other notion of semantic similarity). If that's it, then it's only fair that 4 out 5 reviewers (plus myself) are curious about the motivation for CLIP (also pointing out that it's a strange choice, since it's meant for text-image similarity, and this paper is about text-text similarity). The rebuttal dodged the question every time. It re-iterated that the logits from the entailment classifier are a rather impoverished semantic representation (I, and all reviewers, agree here), but it did never _motivate why CLIP in particular_, as opposed to anything else (in the space of rich text encoders). Because one reviewer explicitly named some BERT-based models, the rebuttal contains an experiment comparing to those, and it looks like those weren't as effective as the proposed CLIP-based approach (whether they were or weren't effective isn't the part that matters, as I explain next). I find this observation uninteresting and insufficient for an answer to the key _why CLIP question_: the authors were probed to motivate their choice, and if I take this table of results for an answer, it basically means 'because we know it works best on the test set'. I don't think that's how it went, I genuinely think the authors performed the experiment to follow up on the feedback. But the lack of interest to engage with the question makes me join the 5 reviewers and perceive the contribution as 'limited'.

If selected for publication, I'd encourage the authors to have the feedback and discussion in this thread reflected in the final draft. The various other clarification questions were good questions, and the rebuttal addressed those well, in my assessment.